



# Deforestation for agriculture leads to soil warming and enhanced litter decomposition in subarctic soils

Tino Peplau[1], Christopher Poeplau[1], Edward Gregorich[2], Julia Schroeder[1]

[1]Thünen Institute of Climate-Smart Agriculture, Bundesallee 68, 38116 Braunschweig, Germany

[2]Ottawa Research and Development Centre, Agriculture and Agri-Food Canada, Ottawa, ON K1A 0C6, Canada

*Correspondence to*: Christopher Poeplau (christopher.poeplau@thuenen.de)

## Abstract

The climate-change induced poleward shift of agriculture could lead to enforced deforestation of subarctic forest. Deforestation alters the microclimate and, thus, soil temperature, which is an important driver of decomposition. The consequences of land-

use change on soil temperature and decomposition in temperature-limited ecosystems is not well understood. In this study, we buried litter bags together with soil temperature loggers at two depths (10 and 50 cm) in native subarctic forest soils and adjacent agricultural land in the Yukon Territory, Canada. A total of 37 plots was established on a wide range of different soils and resampled after two years to quantify the land-use effect on soil temperature and decomposition of fresh organic matter. Average soil temperature over the whole soil profile was $2.1 \pm 1.0°C$ and $2.0 \pm 0.8°C$ higher in cropland and grassland soils

compared to forest soils. Cumulative degree days (the annual sum of daily mean temperatures $> 0°C$) increased significantly by $773 \pm 243$ (cropland) and $670 \pm 285$ (grassland). Litter decomposition was enhanced by $2.0 \pm 10.4\%$ and $7.5 \pm 8.6 \%$ in cropland topsoil and subsoil, compared to forest soils, but no significant difference in decomposition was found between grassland and forest soils. Increased litter decomposition may not be attributed to increased temperature alone, but also to management effects, such as irrigation of croplands. The results suggest that deforestation-driven temperature changes exceed

the soil temperature increase already observed in Canada due to climate change. Deforestation thus amplifies the climate-carbon feedback by increasing soil warming and organic matter decomposition.

## 1. Introduction

The poleward shift of agriculture due to climate change (Franke et al. 2022) will alter the land cover of vast areas in subarctic regions. As the global mean temperature rises, permafrost soils of the boreal forest region thaw (Biskaborn et al. 2019) and

agriculture in high latitudes expands to regions that had previously been less suitable for agriculture (Tchebakova et al. 2011). Climate change warms the Subarctic more strongly than the global average (IPCC 2013). So, subarctic soils are especially prone to SOC loss. Subarctic soils store large amounts of soil organic carbon (SOC) (Hugelius et al. 2014) that are easily decomposable (Mueller et al. 2015). Moreover, the conversion of pristine subarctic forests to agricultural land has been reported to cause large losses of SOC (Grünzweig et al. 2004, Karhu et al. 2011, Peplau et al. 2022), which in turn fosters





climate change. The mechanisms behind deforestation-induced loss of SOC may be manifold and are not understood in detail. This hampers process-based modelling to extrapolate land-use change effects in space and time.

Besides alterations in species composition and net primary productivity, the replacement of forests by open landscapes has a strong impact on the microclimate, particularly on the temperature regime. Due to missing canopy upon deforestation, the ground is exposed to more direct sunlight and air flow is favoured, leading to more variable near-surface temperatures in open

landscapes compared to closed forests (Frenne et al. 2021). The more rapid intra-day temperature changes of the near-surface air have unclear implications for soil temperature. As Lembrechts et al. (2022) showed, there is an offset between air and soil temperature, which depends on the climatic conditions, and soils are around 3.6°C warmer than the air in boreal forests. Surface air temperature may decrease (Lee et al. 2011), but, regardless of the intensity and direction of air temperature changes, little is known about the effects of land-use change on soil temperature. This applies particularly in the context of subarctic

agriculture, since the removal of pristine vegetation and management techniques may have opposing effects on soil temperature. Consequently, potential feedbacks between land-use change, soil temperature and soil organic matter decomposition are also unclear.

Temperature is the most important driver for the decomposition of fresh organic matter (Gregorich et al. 2017), along with moisture (Petraglia et al. 2019) and substrate quality (Fierer et al. 2005). There has been extensive research about the

mechanisms behind the effects of soil warming on organic matter decomposition: at first hand, depolymerization of complex organic structures, microbial enzyme production, sorption processes and aggregate turnover are key for temperature-induced changes in soil organic matter decomposition (Conant et al. 2011). The effect of warming on soil organic matter cycling is indirectly influenced by various site properties, such as evapotranspiration, mineralogy or plant litter chemistry (Davidson et al. 2000) and is, therefore, regionally highly variable (Carey et al. 2016). In subarctic forests, losses of SOC due to accelerated

decomposition exceed the warming-induced gain in SOC due to enhanced net primary productivity (Karhu et al. 2010), as the large share of labile SOC is quickly decomposed upon warming (Peplau et al. 2021). Despite a different composition of SOC in grasslands than in forests (Grünzweig et al. 2004), it has been shown that subarctic grasslands are also highly prone to SOC loss upon warming (Poeplau et al. 2017).

The objectives of this study were 1) to quantify changes in soil temperature when subarctic forest is converted into agricultural

land (i.e. grassland and cropland), 2) to elucidate the influence of various soil properties on such temperature changes and 3) to compare the decomposition of fresh organic matter in forest and agricultural soils. It was hypothesized that the removal of insulating vegetation by deforestation is shifting the soil temperature regime from relatively moderate temperatures in forest soils to more extreme temperatures in agricultural soils with warmer summer and colder winter temperatures in agricultural soils than in forest soils. Furthermore, it was hypothesized that warmer summer temperatures encourage the decomposition of

soil organic matter in agricultural soils compared to forest soils.





## 2. Material and Methods

### 2.1. Research area

A paired-plot litter decomposition experiment was set up in the Yukon Territory in Northwest Canada, at the southern edge of the northern circumpolar permafrost region. The experiment compared litter decomposition and soil temperature in forest and
agricultural land (cropland / market garden, summarized as cropland and / or grassland). Since the Klondike gold rush at the end of the 19th century, the Yukon has an established agricultural sector with farms that are suitable for studying the effects of land-use change from forest to grassland and forest to cropland in the Subarctic. Farms were considered to be suitable for studying the effects of land-use change from forest to grassland or cropland when they (1) originated from forest, (2) were located on mineral soils and (3) had a remaining native forest nearby. Furthermore, both forest and agricultural land needed to
be located on flat terrain with comparable soil properties. This was checked in an auger-based pre-assessment in consultation with the farmers. In total, 15 farms were included in this study and they provided 21 pairs of forest and cropland (n = 12) or forest and grassland (n = 9) (Figure 1).





**Figure 1: Map of the sampling locations and major rivers and settlements of the Yukon. Top right: The Yukon's location (red) within North America (grey).**

## 2.2. Litter decomposition experiment

In order to investigate the effects of land-use change on soil temperature regime and litter decomposition, tea bags and temperature loggers were buried at the chosen farms in summer 2019. Tea bags with green tea (΄Bio Grüner Tee', Paulsen Tee, Fockbeck, Germany, Charge No. 187896FC) as a standard litter material were weighed, tagged and buried at depths of 10 and



50 cm from the soil surface (n=3 per depth), based on the methodology of Keuskamp et al. (2013). Temperature loggers (Tinytag Plus 2 TGP 4017, Gemini Data Loggers Ltd) were buried at the same depths (n=1 per depth) and set up to record the soil temperature every two hours. The tea bags were buried at spots considered as representative of the given plots by placing them approximately 30 cm apart from each other around the temperature loggers. After two years, the tea bags and temperature loggers were dug out in September 2021. The tea bags were cleaned of roots and soil, dried at 60 °C, opened in order to

manually pick out fine roots that grew into the tea bag and weighed again to determine mass loss as a proxy for decomposition. The tea bags from very clayey sites were additionally washed prior to opening to remove clay particles from the tea bag material.

In total, 209 out of 216 tea bags and all 72 temperature loggers were recovered. After downloading the data from the loggers, measurements were checked for plausibility (no abrupt changes that would exceed normal hourly fluctuations) and

completeness (no missing data) before further processing.

## 2.3. Soil parameters

In addition to the burial of tea bags and temperature loggers, soil samples were taken from every plot to characterize the soils of the sites investigated. The sampling was done in summer 2019, at the same time of the burial of the tea bags. Details about the soil sampling and laboratory analyses can be found in Peplau et al. (2022). Soil was sampled from depth increments of 0-

10 cm and 40-60 cm matching the depth of the buried sensors and tea bags. Five field replicates of every depth increment were pooled to a mixed sample and analyzed for organic and inorganic carbon (C) and total N content, $pH_{H2O}$ (ISO 10390), plant available phosphorus (Olsen et al. 1954), SOC fractions (Zimmermann et al. 2007) and texture (Köhn 1929). The soils in the research area were Cambisols and Cryosols (Jones et al. 2009) with pH values between 5.5 and 8.9 (mean: 7.4) and clay contents between 49 and 578 g kg$^{-1}$ (mean: 178 g kg$^{-1}$). Soil parameters and values of SOC stocks were obtained from an

earlier study at the same sites (Peplau et al. 2022). In this earlier study, soils were sampled from 0-80 cm, with depth increments of 0-10 cm, 10-20 cm, 20-40 cm, 40-60 cm and 60-80 cm. The organic C was measured with an elemental analyser (LECO TruMac CN). To distinguish between organic C and inorganic C, samples with pH > 6.2 were heated in a muffle furnace at 440°C before the measurement.

## 2.4. Statistics

The descriptive variables of annual mean temperature, minimum temperature, maximum temperature, temperature amplitude, number of frost days (i.e. days with a mean temperature lower than 0°C) and cumulative degree days (temperature sum of days with a mean temperature above 0°C) were calculated from the original two-year temperature dataset. The number of frost days and cumulative degree days were divided by 2 to obtain the average of both years.

To test for significant differences in litter decomposition and soil temperature parameters between forest, cropland and

grassland, linear mixed-effects models were used with land-use type as fixed effect and site and depth as random effects, allowing for random intercept. Homoscedasticity, normality of the residuals and linearity of the dataset were given and no





transformation of the data was necessary. Since cropland and grassland soils did not have the identical reference forests, separate models were used for cropland/forest and grassland/forest pairs. After performing the linear mixed-effects models, estimated marginal means were used to obtain pairwise comparisons of all groups of the linear mixed-effects model
(confidence level = 0.95).

In order to identify variables that are driving the decomposition of the buried tea bags, the Pearson's correlation coefficient was calculated separately for the complete dataset and for every land-use type and depth.

All statistical analyses were conducted using R version 4.0.4 (R Core Team 2021) with the packages readxl (Wickham & Bryan 2019), tidyverse (Wickham et al. 2019), dplyr (Wickham et al. 2020), purrr (Henry and Wickham 2020), ggplot2 (Wickham 2016), ggpubr (Kassambra 2020), ggthemes (Arnold 2021), ggpmisc (Aphalo 2021), corrplot (Wei and Simko 2021), lme4 (Bates et al. 2015), lmerTest (Kuznetsova et al. 2017), multcomp (Horthon et al. 2008), multcompView (Graves et al. 2019) and emmeans (Lenth 2021). The level of significance for all statistical analyses was selected as $\alpha = 0.05$. All data used for this study is openly available at DOI 10.5281/zenodo.7219753

## 3. Results

### 3.1. Soil temperature as affected by land use

Forest soils were cooler and had smaller intra-day variations in temperature than cropland and grassland soils (Figure 2). During winter, the soil temperature of all land uses did not exceed 0°C and showed very little short-term variations within a couple of days. With the beginning of spring in April, soil temperature at 10 cm depth increased sharply to above 0°C and short-term variations in temperature became larger. At 50 cm, the spring soil temperature increase was visible, but less pronounced than at 10 cm. The sharp increase in soil temperature to above 0°C was visible at all sites at the same time, independent of how low the soil temperature was beforehand. On average, grasslands soils were 2.2 ± 0.9°C and 1.8 ± 0.5°C warmer than forest soils at 10 and 50 cm and cropland soils were 2.1 ± 1.1°C and 2.0 ± 0.9°C warmer than forest soils at 10 and 50 cm (Figure 3), which was significant with $p < 0.05$. Moreover, significantly larger cumulative degree days indicated warmer soils under agricultural use than under forest. Cumulative degree days at 10 and 50 cm were elevated by 606 ± 222 and 733 ± 338 in grassland soils and by 768 ± 262 and 779 ± 237 in cropland soils. This roughly corresponds to a doubling of cumulative degree days upon land-use change. Detailed information about the soil temperature at every site, land-use type and depth can be found in Table S1.

Significant differences between forest and grassland at both depth increments were found in mean temperature, minimum temperature, maximum temperature, total amplitude and cumulative degree days, but not in the number of frost days (Table 1). The comparison between forest and cropland resulted in significant differences at 10 and 50 cm for mean temperature, maximum temperature, total amplitude, cumulative degree days and frost days. In contrast to grasslands, croplands had no different minimum temperature, but fewer frost days than forests. Temperature differences between forest and grassland were smaller in soils with high clay content, while there was no such correlation observed in cropland soils (Figure 4).





## 3.2. Litter mass loss

We observed significant differences in litter mass loss between cropland and forest soils, but not between grassland and forest soils (Figure 5). In forest-cropland pairs, the mean proportional loss of added litter was lower in forest soils (70 ± 7% and 52 ± 4% at 10 and 50 cm) than in cropland soils (73 ± 8% and 61 ± 8% at 10 and 50 cm). In forest-grassland pairs, mean decomposition of added litter in forest soils was 70 ± 4% and 60 ± 12% at 10 and 50 cm, while it was 67 ± 7% and 53 ± 18% in grassland soils at 10 and 50 cm. Detailed information about mass loss at every site, land use and depth can be found in Table

S1.

## 3.3. Soil properties and microclimate explaining tea mass loss

The correlation between litter mass loss and soil temperature, site characteristics and soil properties strongly differed between agricultural land and forest. In forest topsoils, only minimum temperature was significantly negatively correlated with mass loss of tea, while mass loss in subsoils was significantly correlated with minimum temperature, number of frost days and total

temperature amplitude (Figure 6). In cropland soils, significant correlations were only observed in topsoils. In contrast to forest soils, mass loss in cropland soil was positively correlated with minimum temperature. Furthermore, there was a significant negative correlation between mass loss and silt content. In grassland soils, tea mass loss was only correlated with temperature parameters (mean temperature, maximum temperature, amplitude, number of frost days and cumulative degree days). In contrast to croplands, there was no significant correlation between decomposition and SOC fractions in grassland soils. Across

all land-use types and depths, mass loss correlated significantly with soil temperature parameters, except for mean temperature and number of frost days. Weaker, yet significant, correlations were observed between mass loss and soil organic matter (C and N content as well as SOC fractions). This was not observed when separating the sample set into the different land-use types and depths, except for forest subsoils (soil organic matter parameters) and grassland subsoils (temperature parameters). Besides elevated mean and maximum temperature, which may be biased by single extreme values, cumulated degree days also

increased in cropland and grassland soils, compared to forest soils. This increase had a highly significant effect on litter decomposition (Figure 7) ($p < 0.001$). Furthermore, there was a good correlation between SOC stocks and mean soil temperature in forest soils with higher SOC stocks in colder soils, something that was not observed in agricultural land (Figure 8).





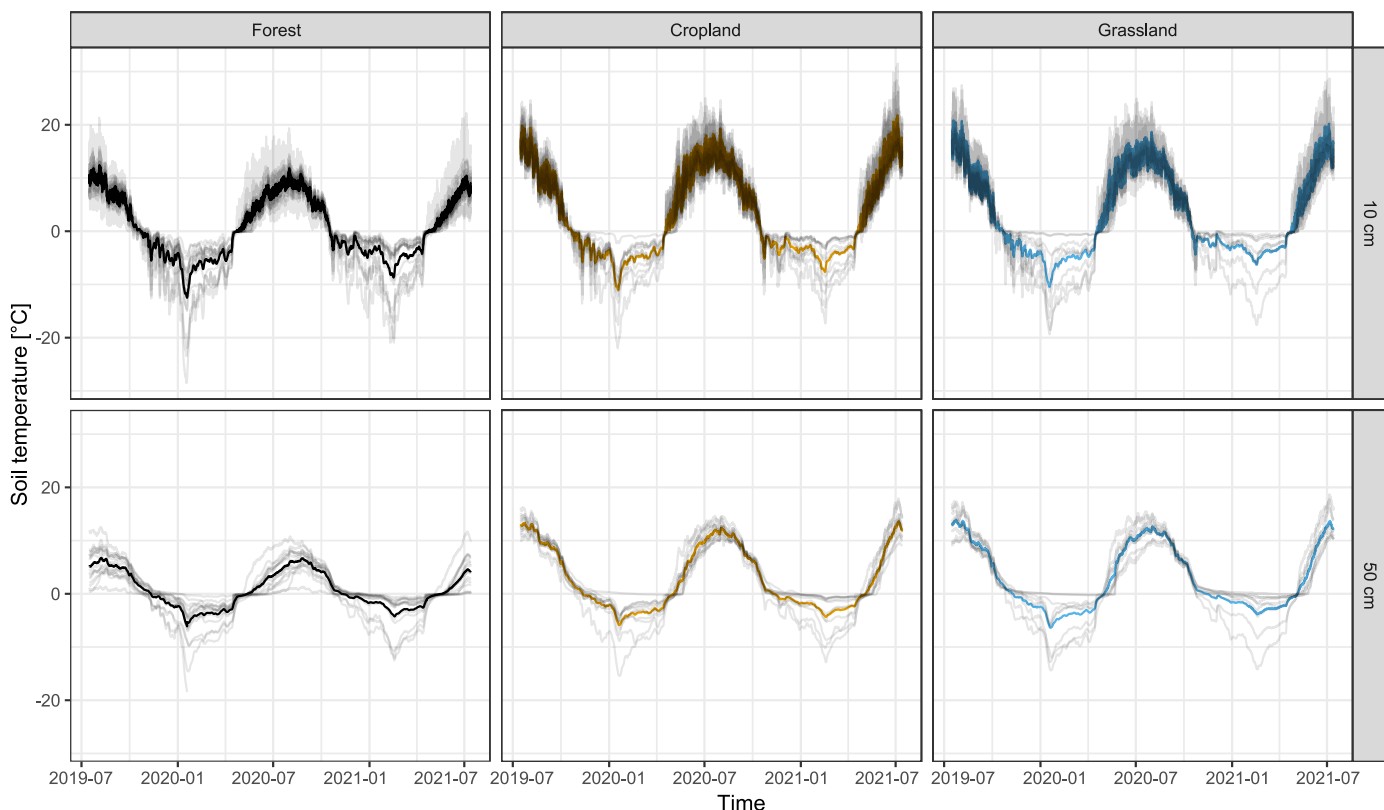

**Figure 2: Soil temperature profile in Forest, Cropland and Grassland at 10 and 50 cm. Grey lines show the temperature of the individual sites; coloured lines indicate average temperatures.**




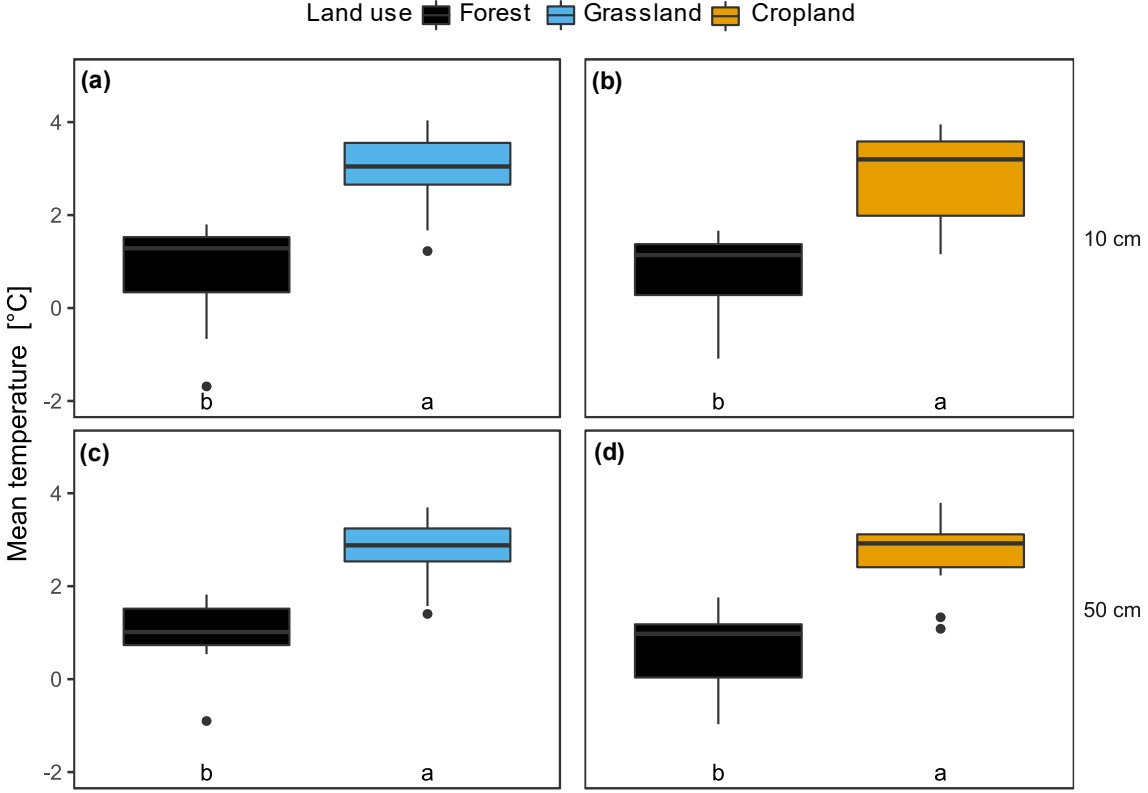

**Figure 3: Tukey-style boxplot of the mean temperature in grassland/forest pairs (n = 9) (a+c) and cropland/forest pairs (n = 12) (b+c) at 10 (a+b) and 50 cm (c+d) depth. Different letters at the bottom of each panel indicate statistically significant differences**



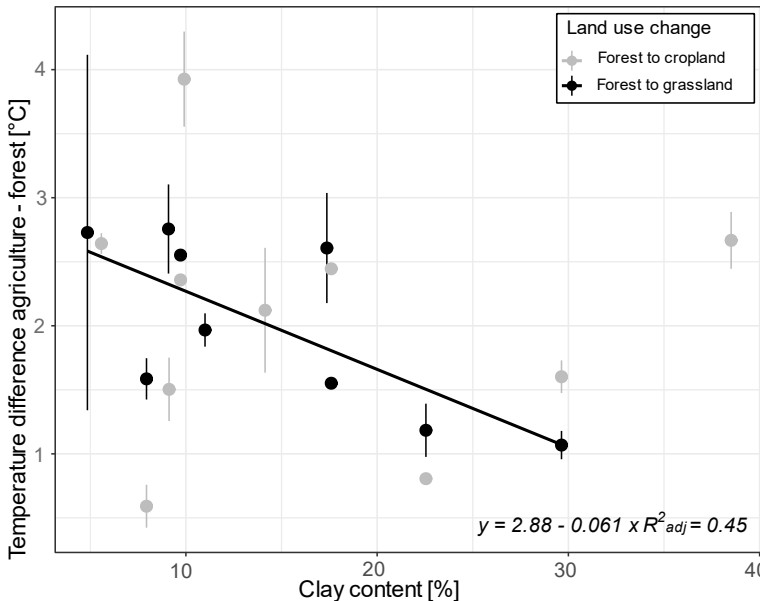


**Figure 4: Relationship between clay content [%] and soil temperature difference between agricultural land and forest. Points indicate mean values between 10 and 50 cm depth; vertical lines indicate standard deviation of the mean temperature between 10 and 50 cm.**



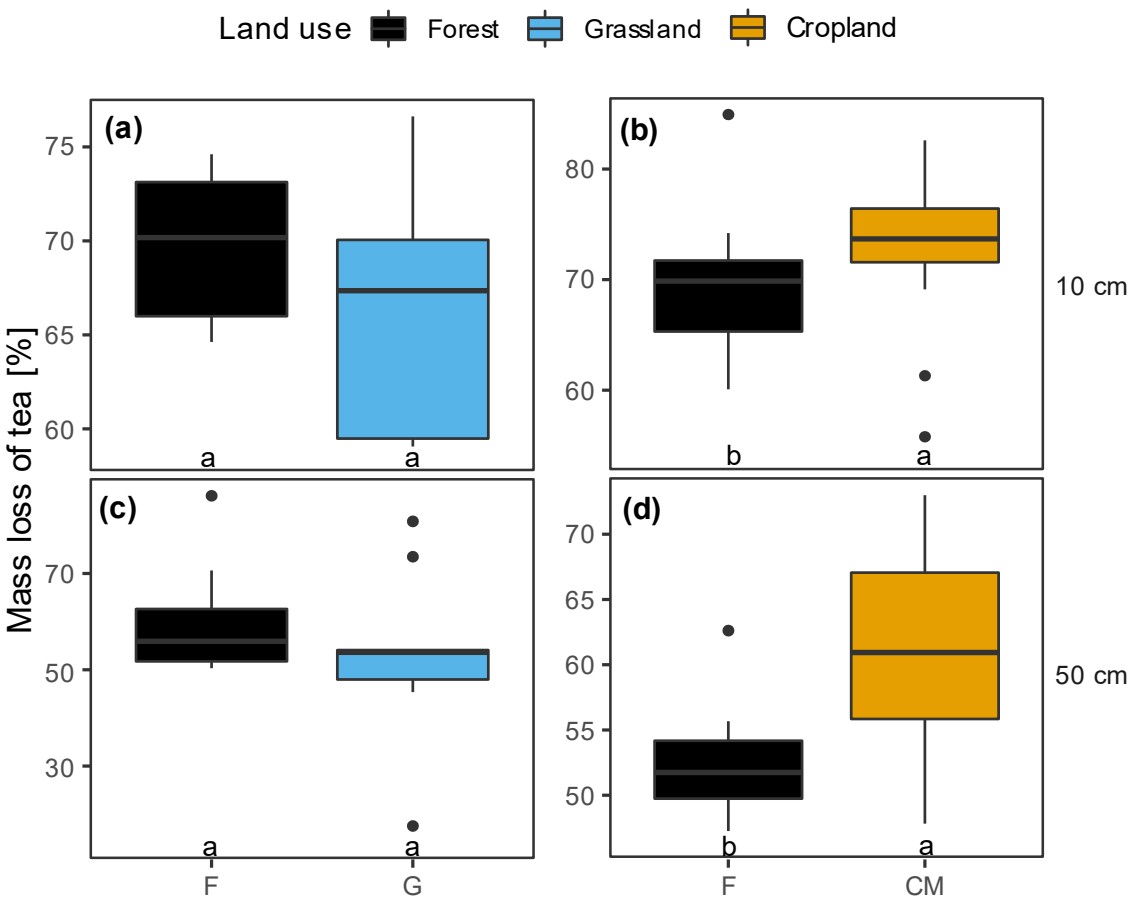

**Figure 5: Tukey-style boxplot, comparing mean decomposition of the buried tea bags in grassland/forest pairs (n = 9) (a+c) and cropland/forest pairs (n = 12) (b+d) at 10 (a+b) and 50 cm (c+d) depth. Different letters at the bottom of each panel indicate statisti**





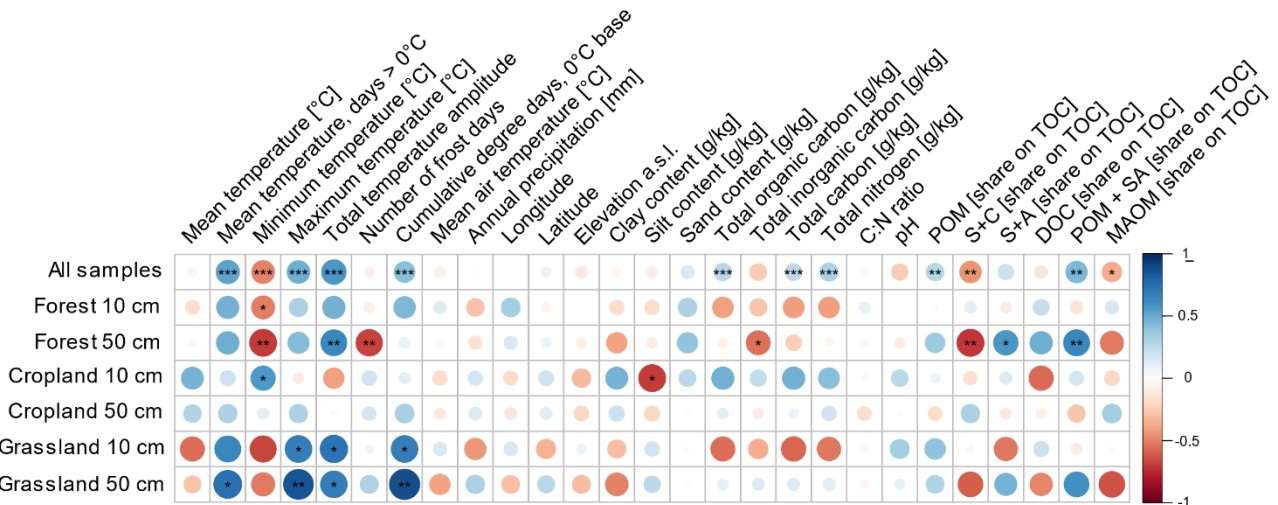

**Figure 6: Correlogram of the Pearson's correlation coefficient, showing the correlation of mean litter decomposition and the most important soil temperature parameters, site characteristics and soil properties. The size and colour of the points represent the dir**


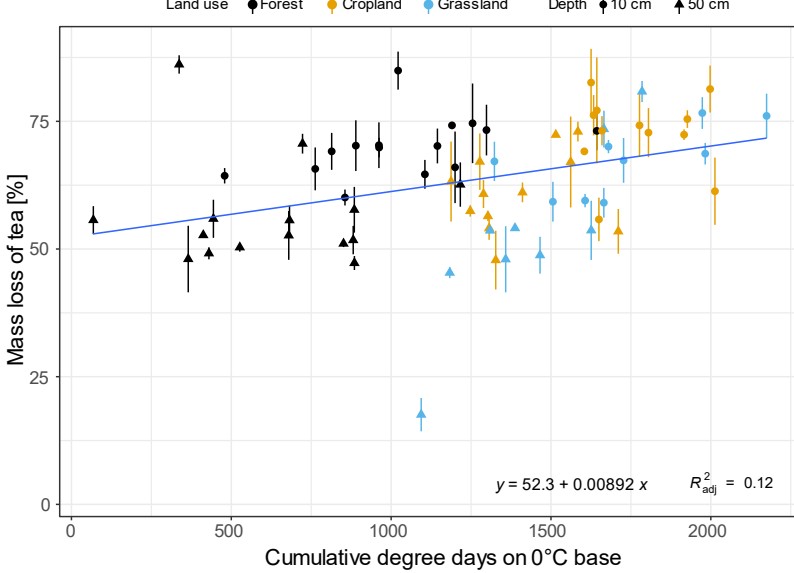

**Figure 7: Point range and regression line (p < 0.05) of tea decomposition over cumulative degree days at 10 cm depth (circles) and 50 cm depth (triangles). Shapes indicate mean values; vertical lines indicate standard deviation of the decomposition. The regressi**





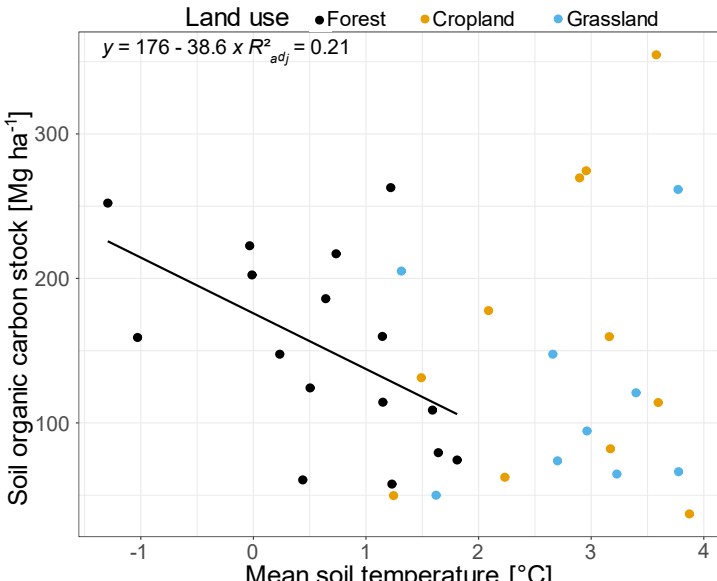

**Figure 8: Soil organic carbon (SOC) stocks between 0-80 cm depth [Mg/ha] and mean soil temperature [°C] of forest, grassland and cropland soils with a significant (p < 0.05) correlation (solid line) between SOC stocks and soil temperature in forest soils. Mean temperature was calculated from soil temperatures at 10 and 50 cm.**

**Table 1: Mean, minimum and maximum temperature, temperature amplitude, number of frost days and cumulative degree days on 0°C base in forest, grassland and cropland soils. Values are means with standard deviation. Asterisks indicate significant difference (p <**

| Land use change | Land use | Depth [cm] | Mean temperature [°C] | Minimum temperature [°C] | Maximum temperature [°C] | Total amplitude [°C] | Frost days | Cumulative degree days |
|---|---|---|---|---|---|---|---|---|
| Forest to grassland | Forest | 10 | 0.7 ± 1.2 | -14.3 ± 8.3 | 14.1 ± 3.8 | 28.4 ± 11.3 | 188.9 ± 9.1 | 1131.6 ± 268.3 |
| | | 50 | 1.0 ± 0.8 | -7.5 ± 6.1 | 8.4 ± 2.2 | 15.9 ± 7.8 | 168.2 ± 51.3 | 697.4 ± 288.8 |
| | Grassland | 10 | **2.9 ± 0.9\*** | **-10.9 ± 6.7\*** | **22.5 ± 4.6\*** | **33.4 ± 10.5\*** | 189.8 ± 6.4 | **1737.6 ± 264.0\*** |
| | | 50 | **2.7 ± 0.8\*** | **-7.0 ± 5.5\*** | **14.6 ± 2.6\*** | **21.6 ± 7.5\*** | 154.9 ± 31.3 | **1430.5 ± 228.1\*** |
| Forest to cropland | Forest | 10 | 0.8 ± 0.8 | -12.1 ± 5.1 | 12.9 ± 3.6 | 24.9 ± 8.3 | 195.4 ± 12.5 | 1011.2± 290.1 |
| | | 50 | 0.6 ± 0.8 | -6.2 ± 3.8 | 7.2 ± 2.8 | 13.4 ± 6.2 | 197.4 ± 23.5 | 636.3 ± 317.7 |
| | Cropland | 10 | **2.9 ± 0.9\*** | -12.0 ± 4.4 | **23.6 ± 4.3\*** | **35.6 ± 7.3\*** | **187.8 ± 6.4\*** | **1771.0 ± 156.0\*** |
| | | 50 | **2.7 ± 0.8\*** | -6.4 ± 3.7 | **14.5 ± 1.7\*** | **20.9 ± 4.7\*** | **168.7 ± 18.7\*** | **1393.3 ± 162.0\*** |



## 4. Discussion

### 4.1. Land-use change alters soil temperature in subarctic soils

The higher temperatures, cumulative degree days and greater amplitude that we measured in grassland and cropland soils supported our hypothesis that deforestation is shifting the soil temperature regimes from a moderate temperature amplitude, with relatively low summer temperatures in forest soils, to more extreme amplitudes in agricultural soils, with particularly warm summer temperatures. The observed soil warming upon deforestation is in line with results from earlier studies, reporting temperature increases of 2.0°C in the tropics (Jiménez et al. 2007), between 2.5 and 3.3°C in the temperate zone (Morecroft et

al. 1998) and up to 5.0°C, during summer, in boreal Alaska (Grünzweig et al. 2003). Similar to our results, forest soils in all of the three studies mentioned were on average cooler in summer compared to agricultural land, which is due to shading by the forest canopy. During summer, we observed that forest soils were on average 4.0°C cooler in topsoil and subsoil than croplands and 3.8 and 4.2°C cooler than grasslands, which is slightly less than observed by Grünzweig et al. (2003). Also, cumulative degree days increased between 600 and 800, which is almost a doubling of cumulative degree days upon land-use

change. This is a similar increase as reported by Grünzweig et al. (2003), which was between 500 and 650 annually. As shown in a modelling study for all of Canada, warmer winter soil temperatures can be related to thicker snow cover in deforested land compared to forest (Zhang et al. 2005). Snow cover, along with vegetation, is the most important factor determining soil temperature patterns (Qian et al. 2011, Zhang et al. 2005). Warmer winter soil temperatures in agricultural land compared to forest as a consequence of thicker snow cover on open land than on forests were also reported by Grünzweig et al. (2003).

Agricultural soils of the Yukon were equally cold (cropland) and slightly warmer (grassland) than the forest soils during winter. This emphasizes the importance of vegetation for the soil temperature. Snow cover on bare soil might insulate the soil in a similar way to natural forest vegetation, but the combination of dense grasses and overlaying snow cover adds an additional insulation effect.

The temperature difference between forest and agricultural land appeared to be influenced not only by insulation of the soil by

vegetation or snow but by inherent soil thermal properties, which are regulated by soil texture. Clayey grassland sites had smaller temperature differences upon deforestation than sandy sites. This can be related to the differences in the thermal properties of air, water and different minerals with clayey sites having the lowest and sandy sites having the highest thermal conductivity (Dong et al. 2015). Overall, clayey soils have a larger pore volume and are, therefore, more buffered thermally than sandy soils, if the pore space is not water filled but contains a lot of air. Under wet conditions, heat exchange between the

soil and the atmosphere is increased and soils cool down more strongly than under dry conditions. In cropland soils, no relationship between soil texture and temperature was supported statistically. Since croplands in the Yukon are irrigated regularly, the thermal insulation of the soils might be reduced at all cropland sites, independent of soil texture, which was not the case in unirrigated grassland sites. Moreover, the effects of soil properties on the soil temperature regime might be masked by differences in vegetation.




As we have shown, land-use change has a soil warming effect of around 2.1°C. Due to climate change, Canadian soils warmed by 0.6°C during the 20th century, with regional differences between -2 and +5°C, underlining the great importance of spatially distributed soil temperature measurements (Zhang et al. 2005). Climate change related alterations in the temperature of Canadian soils have been observed to be greatest in spring (0.26 – 0.30°C per decade since the 1950s), while winter soil temperatures have not changed significantly (Qiang et al. 2011). In contrast, air temperatures in Canada have increased most

strongly in winter (2.3°C between 1950 and 2010) and less so in spring (1.7°C between 1950 and 2010). The annual mean air temperature increased by 1.5°C between 1950 and 2010 or 0.25°C per decade (Vincent et al. 2012), which is slightly less than the increase in soil temperature within the same time. Our results imply that land-use change from pristine forest to agriculture exceeds the effect of climate change on soil temperature and is particularly strong during summer, when biological activity is highest in subarctic ecosystems. Common models of SOC turnover are fed by air temperature instead of soil temperature

(Balesdent et al. 2018, Kaczynski et al. 2017, Crowther et al. 2016). Thus, one of the most important drivers of microbial activity and SOC mineralization, that is temperature, is assumed to be independent of the vegetation cover. The present study highlights that this might be a severe shortcoming in such models, which often fail to capture land-use change effects (Boysen et al. 2021, Gottschalk et al. 2010).

### 4.2. Litter decomposition and its implications for SOC dynamics in subarctic soils

It was hypothesized that warmer summer temperatures foster the decomposition of fresh soil organic matter in agricultural soils compared to forest soils. Indeed, there was a greater mass loss of tea in cropland soils than in forest soils. Particularly in subsoils, mass loss was around 8.7% higher in croplands than in forests, while it was 2.9% in topsoil, which might be related to the fact that the temperature effect was more masked by agricultural management in the topsoil. However, the hypothesis must be rejected for grassland soils, since there was no significant difference in litter decomposition between forest and

grassland soils, despite warmer soil temperatures in grasslands than in forests. This might suggest that not only did the deforestation induce soil warming but also agricultural management controlled litter decomposition. In a global study, Djukic et al. (2018) reported that precipitation is the most important climatic factor for litter decomposition and temperature appeared to be less important. However, their study was conducted only during summer, where water availability, and not temperature, was the limiting factor. Our results suggest that, in temperature-limited regions, the temperature may play an important role in

litter decomposition, as shown by the relationship between cumulative degree days and litter mass loss. However, missing evidence for this relationship in grassland/forest pairs might underline the importance of water availability as a prerequisite for decomposition. Soil moisture was not quantified in this study, but we can assume that croplands had higher soil moisture than grasslands, as croplands are irrigated regularly due to the dry climate in the research area (on average, 262 mm annual precipitation (Environment Climate Change Canada 2020)) and grasslands remain rainfed. However, litter decomposition

might be underestimated in grassland plots, since there were more fine roots potentially growing into the tea bags, which might not have been removed entirely before weighing.





Various studies have reported losses of SOC after land-use change (Grünzweig et al. 2003, Grünzweig et al. 2004, Guo and Gifford 2002, Wei et al. 2014, Poeplau 2011). A certain proportion of these losses can be assigned to deforestation-induced removal of the uppermost soil layers, including litter and parts of the topsoil (Grünzweig et al. 2003). C input quantity (Luo et al. 2017) and quality (Cotrufo et al. 2019) as well as frequent soil disturbances and changes in aggregate stability (Six et al. 2000) can add to the land-use change driven alterations in SOC stocks. Here, we were able to show that microclimatic changes and their effect on litter decomposition are another relevant driver of SOC stock change after deforestation. Under natural conditions, as represented by the forest sites, SOC stocks were related to mean soil temperature to some extent. The coldest forest soils stored significantly more SOC than the warmest forest soils, with a linear decrease in SOC of 38.6 ± 17.2 Mg C ha$^{-1}$ per °C ($p < 0.05$) warming (Figure 8). This slope is one order of magnitude higher than values observed in warming experiments (Peplau et al. 2021: 1.9 Mg C ha$^{-1}$ per °C, Verbrigghe et al. 2022: 2.8 Mg C ha$^{-1}$ per °C). This might indicate that the observed range in forest SOC stocks cannot solely be explained by a direct temperature effect. Instead, it has been observed that the coldest sites, which were also characterized by shallow permafrost (detectable ground ice in summer within the upper 50 cm), were rather wet sites with thick, C-rich A horizons. Farmers reported that the waterlogging ceased after deforestation, i.e. with the deepening of the permafrost layer (Peplau et al. 2022). Despite a high general variability in SOC stocks across forest sites, our data suggests that, in the presence of permafrost, warming might have a much more severe effect than in non-permafrost soils. This is because water infiltration is hampered by underlying ice layers and soils remain wetter during summer than sites without permafrost. The linear relationship between soil temperature and SOC stocks was not apparent in agricultural soils, although SOC stocks were reduced significantly in cropland soils (Peplau et al. 2022). The decoupling of SOC stocks from soil temperature in agricultural soils shows that the natural, climate-driven balance between C input, mineralization and C storage is heavily disturbed by agricultural activity. Depending on agricultural practices (i.e. cultivated crops, soil management, irrigation), the amount and quality of C input varies greatly from natural habitats and also between sites. This makes the quantification of land-use change induced alterations of C mineralization more complicated. Given that land-use change will increase soil temperatures in subarctic soils, the future spread of agriculture on permafrost soils may therefore additionally accelerate the climate-carbon feedback, causing more SOC loss than already caused by climate change and the warming of pristine forests. The observed different effects of land-use change on soil temperature and decomposition in croplands and grasslands indicate that changes in the soil water regime might be essential for prospective land-use carbon feedbacks. Since warmed soils have a deeper (or entirely thawed) active layer, soils naturally dry upon warming due to drainage effects (Andresen et al. 2020).

## 5. Conclusion

The aim of this study was to quantify the effect of land-use change from subarctic forest to agricultural land on soil temperature and decomposition of fresh soil organic matter. Using tea as a standardized litter material which is easy to compare under varying site properties allowed us to couple soil temperature as influenced by land-use type with litter decomposition in diverse

soils under subarctic agriculture. Overall, the effect of land-use change on soil temperature exceeded the effect of past climate
change and thus strongly enhances the climate-carbon feedback. Deforestation resulted in soil warming, but the consequences for litter decomposition depended on the subsequent management. Cropland soils, which are more often disturbed by field operations and irrigation, had a greater decomposition of fresh organic matter than forests. This was not the case in grasslands, even though they exhibited a greater difference in soil temperature. Therefore, future climate mitigation strategies and modelling efforts need to consider the effect of land cover on soil temperature changes, additional to air temperature changes.

**Data availability**

All data used for this study is openly available at DOI 10.5281/zenodo.7219753

**Author contribution**

C.P. and J.S. designed the tea bag and temperature experiments and were responsible for the field setup. J.S. and T.P. finalized the experiments. T.P. was responsible for writing the original draft, including figures and statistical analysis, and all co-authors
contributed by reviewing and editing. E.G. provided additional language editing.

**Competing interests**

The authors declare that they have no conflicts of interest.

**Acknowledgements**

We would like to thank all the farmers who participated in this study and allowed us to use their land for our experiments. We
are also grateful to Yukon's First Nations who granted us permission to study their traditional land, without which this study would not have been possible. This study was part of the project 'Braking the Ice', funded by the German Research Foundation, grant number 401106790. Funding for E.G. was provided by the Science & Technology Branch of Agriculture & Agri-Food Canada (Project J-001756 'Biological Soil Carbon Stabilization').

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
