# Peer review of "Deforestation for agriculture leads to soil warming and enhanced litter decomposition in subarctic soils"

_EGUsphere, 2022_

## Author Response (AR1)

Dear editor and anonymous reviewers,

thank you for your effort in revising our manuscript and for the opportunity to hand in an improved version. We carefully considered all of your comments and made changes according to them. We provided more detail to the material & methods section, including a more detailed description of the experimental design and farming practices and we revised the statistics carefully. In the discussion section, we added more thoughts on the potential factors that may have affected our results and we discussed the utilization of tea bags in terms of litter decomposition studies in general. Overall, we think that the manuscript improved a lot and we would be happy if you consider it for publication.

Kind regards,

Tino Peplau

**Reviewer Comments #1**

The manuscript "Deforestation for agriculture leads to soil warming and enhanced litter decomposition in subarctic soils" is well written and offers valuable soil temperature data concerning land use change. Further, it highlights the possible importance of other management-related effects on litter decomposition rates.

I have some comments regarding the data analyses and reporting: I am not sure that it is correct to treat "depth" as a random effect. This is indeed an effect that you should be interested in and including it as a fixed effect also allows you to explore interactions. Further, I suggest presenting exact p-values rather than <0.05, <0.001 etc. Throughout the text the authors also present mean values ± some value of their variation. However, it is not always clearly stated whether this is standard deviation or error. See for example section 3.1 and 3.2.

Answer: Thank you for your useful comments regarding the statistical analyses. Although we initially did not consider depth to be a factor of major interest, we agree that treating depth as fixed effect adds a higher value to the analyses. However, no statistically significant effect was found between land-use type and depth, neither for the comparison between forest and cropland, nor between forest and grassland. We revised the manuscript to give exact p-values and included a statement on the mean values ± standard deviation to L130.

The material and methods section would likewise benefit from a bit more clarity. For instance, the experimental set-up is not easy to follow – here a figure might help. There are 21 pairs but only 15 farms. Does this mean that some farms had both cropland and grassland? If so, did they share the same forest? Did any farm have two croplands or grasslands? See specific comments relating to other parts of the method section below.

Answer: Yes, at some of the farms we sampled triplets of land-use types, including forest as well as both cropland and grassland. The supplementary Material provides information on which pairs can be found at each farm and how the teabags and temperature loggers were buried. At one site (LR), we sampled two different cropland plots, with the same forest plot as reference. However, we see that this is not clear from the beginning, and we refer to the supplement in section 2.1. Research area. Additionally, we changed L. 721ff, which now reads as follows: "15 farms, covering pairs of forest and cropland, forest and grassland and triplets with forest, cropland and grassland,

were included in this study. These farms provided 21 pairs of forest and cropland (n = 12) or forest and grassland (n = 9) (Figure 1)."

Otherwise the text is well structured and addresses the three objectives that it set out to answer. My opinion is that the manuscript can be published after considering the comments stated below.

Answer: Thank you for your positive feedback on our manuscript. We appreciate your comments and have considered the points you raised.

**Specific comments**:

Line 55-60: No hypotheses were given in connection to the second objective. Did you already have some ideas about which soil properties might affect soil temperature beforehand, or was this part exploratory?

Answer: We decided to keep the second objective more exploratory, as the connection between soil properties and soil temperature are very complex and depend on a lot of other variables such as vegetation cover or relief. Therefore, formulating a clear hypothesis for this objective would have been more of a guess than an actual hypothesis.

Line 65-69: it would be useful to have slightly more detailed site descriptions, e.g. concerning vegetation. What kind of native forests do we find in this area? Which types of croplands are you investigating? I am a bit unsure what the term "market garden" implies. It becomes especially important to provide this information when you then go on to emphasize the importance of vegetation for soil temperature (Line 216).

Answer: Thank you for this comment. We see that the term "market garden" might be confusing in this context and omitted it. Instead, we described the types of cropland in more detail. We summarized (and cited) our earlier study (Peplau et al. 2022) in which we described the native forests and the different types of farmland and their management in detail. L. 74ff now reads: "As described in detail in Peplau et al. 2022, forests in the research area were mixed-wood forests of the boreal cordillera ecoregion, croplands were small scale fields with grains, potatoes or vegetables, herbs and greens whereas grasslands were used as pasture for livestock grazing or for hay production."

Line 69: What is meant by "nearby"? You could consider providing a number stating average within-pair distance (and between-pair distance, see comment on figure 1 below).

Answer: We agree that „nearby" might be misleading, as the forest and the agricultural land was usually directly adjacent. We revised the wording and indicated a within-plots distance (roughly between 20 and 100 m) in section 2.1. Research area (L 69).

Figure 1: Even though sampling points are given in figure 1, it is difficult to evaluate the between-pair distance (also since some farms have more than one pair).

Answer: Thank you for this comment. We revised the wording in the text (L. 67): "Farms were considered to be suitable when they [...] (3) had a remaining native forest adjacent to the agricultural land, i.e. within a distance of approximately 100 m." With regards to details on the paired plots per site we referred to the supplemental material in L 73ff.

Line 70: It is not a simple task choosing comparable site pairs; you must have put a lot of effort into this. Maybe you could elaborate on what is meant by "comparable soil properties (…) checked in an auger-based pre-assessment", i.e. what were the soil criteria for the pairs. Soil type, soil texture etc.?

Answer: We agree that there should be given more detail. Beside soil types, we checked for soil texture and the proportion of rock fragments, bands of organic material as well as the visibility of hydromorphic properties. L 70 now reads: This was checked in an auger-based pre-assessment in consultation with the farmers, focusing on soil texture and the proportion of rock fragments, bands of organic material as well as on the visibility of hydromorphic properties.

Line 79: Regarding the chosen depths of teabag deployment. You are using relatively fresh leaf litter material, which is normally deposited on top on the soil. Did you consider burying the "topsoil-tea" at a shallower depth? Since "the soil was sampled from depth increments 0-10 cm and 40-60 cm" and the "subsoil-teabags" were placed at 50 cm, it would intuitively have made more sense to bury the "topsoil teabags" at a depth of 5 cm (or shallower). Were you trying to avoid the teabags being dug up (e.g. by shallow tillage in the croplands)? Can you provide a rationale for choosing these depths.

Answer: We agree that the topsoil position does not ideally correspond to the first sampling depth (0-10 cm). But in fact, the soil was also sampled in 10-20 cm and many of the analyses (e.g. soil texture and fractions) were done only in 10-20 cm. The rationale was basically to represent potential litter decomposition in top- and subsoil. Topsoils, especially in agricultural soils, had a thickness of more than 10 cm, so the temperature loggers were placed in 10 cm depth to get an average topsoil temperature. Teabags were placed in the very same depth to relate both to each other. Also, at least in agricultural soils, fresh litter is regularly incorporated into a certain depth (something between 10 and 30 cm). To be able to compare the effect of microclimate on potential litter decomposition between agricultural and native soils, teabags were placed likewise in 10cm depth in forest soils.

Line 84-87: It is very nice that you have been so careful when weighing the tea i.e. washing off mineral soil particles and opening the bags to take out fine roots, but you must then also have obtained the (average?) weight of the teabag-bag (to be able to subtract this weight from the initial weights to gain estimates of start-weights including only tea). Maybe it could be nice to mention how you obtained this value (e.g. obtained from "fresh" bags; how many?).

Answer: Before the start of the experiment, we opened a total of 12 fresh teabags and weighed only the bag material (including nylon bag, thread and tag) to know the average weight of the nylon bags. This value was used as bag weight for all buried tea bags. We now added the following sentence to the manuscript (L. 85): "Before the start of the experiment, 12 tea bags have been opened and weighed to determine the weight of the bag material without the tea."

Line 110: I am not sure it is justified using "depth" as a random effect here. You may choose an effect to be random if you are not interested in the effect itself, but only want to "control" for it. Another rule of thumb is that you need many levels of the effect to make it random (i.e. more than two). Neither of which is the case here.

Answer: We agree on your comment to treat "depth" as fixed effect. Please see our answer to your general comments on this point.

Line 133: It would be useful to provide the actual p-value since alpha-levels are somewhat arbitrary. Then the reader would have a better idea about the strength of the evidence.

Answer: Agreed. We added the actual p-values throughout the manuscript.

Line 138-140/Table 1: It is great that you provide effect-sizes and uncertainties, but you could also consider supplying p-values, to help the reader asses the strength of the evidence more easily.

Answer: Agreed. We added the actual p-values throughout the manuscript.

Line 165/figure7: You state p<0.001 in the text and p<0.05 in the figure. Why not provide the actual p-value? Also, even though you are correct that the effect is "highly significant", I would be careful how I report this relationship with an R2(adj) of 0.12. I suggest that you add a line acknowledging the low R2-value.

Answer: Agreed. We added the actual p-values throughout the manuscript.

Line 248: I got a little unsure here. Can you be more specific about how/which type of agricultural management masks this temperature effect in the topsoil? In this connection, it would also be useful to describe whether any tillage or fertilizer is applied to the croplands (e.g. in the method section). Further, I am curious to know if the specific areas in the cropland where the temperature loggers and teabags were buried were treated differently than the rest of the cropland soils, e.g. if farmers avoided tillage or planting of crops directly inside the study plots?

Answer: We agree that more detail is needed here. As we tried to keep it brief, we omitted the information about agricultural management but it might help the reader to understand the processes involved. We added some additional information to section 2.1. (Research area) and referred to our earlier study for more details. Furthermore, we extended the sentence in L 266: "Particularly in subsoils, mass loss was around 8.7% higher in croplands than in forests, while it was 2.9% in topsoil, which might be related to the fact that the temperature effect was more masked by agricultural management in the topsoil, where tilling, harvesting and other practices leaded to a regular disturbance of the soil." Indeed, plots where the teabags were buried were not tilled during the two years of burial. Apart from that the plots were treated similar to the rest of the field site, which was possible since due to small scale systems most work on field is done manually anyway.

Line 258: It is a weakness that the study did not obtain soil moisture data to support claims about irrigation effects on litter decomposition, but you could strengthen your argument if you had some sources to back up the assumption that "croplands are irrigated regularly", e.g. personal communication with farmers or similar.

Answer: Thank you for your input. Indeed, the farmers told us about their fields being irrigated, which was also published in Peplau et al. 2022. We now extended the sentence in Line 280 by "[…],according to a farmers questionnaire conducted in Peplau et al. 2022"

**Technical corrections**:

Line 11: "litter bags", consider specifying: "tea bags".

Answer: We changed "litter bags" to "tea bags".

Line 78: For how long were the bags buried? I would prefer exact start and end dates. Currently It says that they were buried in summer 2019 and retrieved in September 2021.

Answer: We now added a table to the supplement.

Line 78-80: A slightly picky comment. It is good that you refer to the methodology of Keuskamp et al. 2013 when using teabags, even if your methods deviate substantially from their methodology. However, it might (?) confuse some readers that you refer to "green tea" in the same sentence when it is in fact not identical to the tea used in the Teabag Index. I suggest that you are more clear about it being a modified method, i.e. deviating from the cited method by using different depths, different incubation time and different tea; why you were not able to calculate decomposition rate constant (k) and stabilization factor (S).

Answer: Thank you for your comment. We think that the reference should be kept in the text as our experiment was largely inspired by this. We agree that more detail about how we deviated from the original methodology is needed and we added the following to L 80ff: "This methodology is based on the work of Keuskamp et al. (2013), but slightly modified in terms of burial depth (10 and 50 cm instead of 5 cm), choice of tea (only green tea instead of green and rooibos tea) and duration of burial (two years instead of three months), as these modifications were necessary to meet the objectives of our study."

Line 102: Do you have a reference for this method to distinguish organic and inorganic

Answer: We added a reference (Byers et al. 1978).

Figure 3/5/7 + Table 1: Figure text is missing. But I assume this to be an artifact of layouting. Maybe also doublecheck the other figures.

Answer: All captions were included in the initial submission. We will make sure to have everything included when uploading the next version of the manuscript. Thanks for noticing!

Figure 5: I suggest using the same scale on the y-axes for easier comparisons.

Answer: We see that different scales are somewhat problematic as they could confuse the reader and can be misleading. Nonetheless, we decided to have different scales on the y-axes since the different thickness of the boxes, especially concerning 5c and 5d, make it hard to properly distinguish all of the details of the boxes otherwise

**Reviewer Comments #2**

This study aims to assess the effects of land use change on litter decomposition. To do so, the authors conducted a field incubation experiment by burying tea bags at two depths (10 and 50 cm) in a total of 37 plots. After two years of incubation, the tea bags were harvested to measure mass loss. This loss was then correlated to a series of climate and soil variables. In general, this paper is easy to follow and collected a potentially useful data set. However, I have some general comments about the scientific significance of the experimental design, statistical assessment of the data, and implication of the results, which would need to be properly addressed before publication.

Experimental design. This paper mainly focuses on litter decomposition. It is difficult to understand why tea bags (which are tree leaves) were buried into the soil rather than put on the soil surface. In the soil, particularly in deeper layers such as 50 cm, most plant litter would be root exudates or dead roots. The tea bag seems cannot represent this litter type. More importantly, with the proceeding of tea bag decomposition, it will get mixed up with soils. Together with other confounding factors, it will certainly overestimate mass loss when the harvest bags were washed to estimate remaining mass. So, the data collected in this study would be suffer from large uncertainties, which should be corrected or discussed at least.

Answer: Thank you for your thoughts about the experimental design. Green tea is a commonly used material for litter decomposition studies, which is why we decided to choose it for our study. Tea bags have the advantage over other litter bags that they are standardized and easy to obtain. It is true, that subsoils would not necessarily see such high amounts of leave litter (or leave litter at all), but here we used it as a proxy of "potential litter decomposition" as a function of pedoclimatic conditions (which are strongly altered by land use change). In this way, we are able to show that also decomposition rates in the subsoil (of a standardized material) are affected by land use change. This would have been much more difficult to experimentally test with root exudates. In this sense, we think that using teabags is a valid approach, although measured decomposition does not tell anything about the turnover of carbon that naturally reaches the subsoil. We clarified this in the following statement at the beginning of discussion section 4.2.: "Using tea bags instead of naturally occurring litter had the advantage that the material was standardized, eliminating effects on decomposition caused by differences in litter material between sites or within the soil profile. However, the decomposition values obtained can only be interpreted as potential litter decomposition and not as the actual decomposition.". As noted in the discussion (L 280) we are aware of uncertainties in the dataset and tea loss might not have been accurately determined in all cases due to roots growing into the tea bag which might not have been fully removed by the hand-picking process. Mixing of soil and tea only happened at the two most clayey sites, which is why these bags were additionally washed. During the root-picking afterwards, no soil particles were found in the tea bags. However, since a correction of these uncertainties seems not feasible we extended the discussion about that in section 4.2 (L. 282): "Furthermore, despite no soil particles were visible within the teabags after opening the bags after burial, there might be a slight underestimation of the decomposition caused by small clay particles sticking between the remaining tea leaves.".

Statistical assessment. I agree with the comments from the other reviewer about the statistic assessment. The paper indeed collected a comprehensive data set (if we do not consider the experimental design). Except the basic ANOVA and correlation analyses, I suggest the author to further explore the data using more advanced statistic techniques to derive the controls on the mass loss and the relevant potential interactions and potential nonlinear relationships. As pointed out by the other reviewer, the current assessment based on those ANOVA test can be improved. For example, for the ANOVA in Figures 3 and 5, why not conduct a pair-wise comparison? Burying depth can also be treated as a factor beside land use, i.e., two-factor ANOVA. For those regression assessment, mixed-effects linear models also can be tried to assess the multilevel structure of the controls.

Answer: Thank you for your ideas about how to improve the statistical assessment. Instead of basic ANOVA, we used a linear mixed effects model to assess the interactions between potential litter decomposition, land-use and depth. According to the reviewers' comments, we now include depth as a fixed effect. Furthermore, we calculated estimated marginal means to compare the different groups of land-use type and depth.

Implication of the results. As the authors did not well explain the scientific rational of burying tea bags to the deep soil (10 cm and 50 cm), it is difficult to connect the results to real implication in the field. It must be highlighted that surface and buried litter will show distinct decomposition behaviors. The results generated by this experiment may only represent a special case of burying leaves into deep soil, which is rarely occur in natural ecosystems (in agricultural systems, this would be common due to tillage). In addition, the soil environment is complex in terms of the controlling factors of litter decomposition. Temperature is only one of those controls. However, this paper provides little information about how temperature interacts with other soil attributes (particularly soil moisture) to regulate litter decomposition (I think the data has the potential to do so). I think that changes in soil moisture regimes would be more important than temperature in controlling litter decomposition in the soil.

Answer: Thank you for your comment. This study aimed to investigate the effect of land-use change on potential litter decomposition, which is a function of pedoclimatic conditions. Pedoclimatic conditions are strongly altered by land-use change and therefore it was hypothesized, that potential litter decomposition is affected by land-use change. Standardized tea leaves litter mass loss (buried to 10cm and 50cm depth) was used as a proxy for the "potential litter decomposition". For clarification, we added a statement that decomposition values are referring to potential decomposition and do not correspond to the actual decomposition of naturally occurring litter (L. 264). In this study, we were able to show that land-use change does indeed affect potential litter decomposition in topsoil but also in subsoil by affecting pedoclimatic conditions (e.g. soil temperature). Unfortunately, the financing of the project did not allow to equip the plots with additional soil moisture loggers. We emphasized the shortcoming of missing soil moisture measurements and added a statement how we accounted for differences in soil moisture between land-use types (L 236: "Measurements of soil moisture have not been conducted in this study, but but since the study region has only little precipitation and farmers irrigate their croplands regularly, it is likely that the irrigated land is on average wetter than the forest (Peplau et al. 2022)." Since no direct measurements of soil moisture were conducted, further interpretations are difficult to justify. Regarding the soil attributes controlling soil temperature, we pointed out that there is an interaction between soil moisture, as influenced by soil texture, and temperature (L 234).